# Investigation into Mining Economic Evaluation Approaches Based on the Rosenblueth Point Estimate Method

**Jiaoqun Li [1], Tong Wu [1], Zengxiang Lu [1,\*] and Saisai Wu [2,\*]**

[1] School of Mining Engineering, Liaoning University of Science and Technology Liaoning, Anshan 114051, China; ljq_as@ustl.edu.cn (J.L.); 2604452468@ustl.edu.cn (T.W.)

[2] School of Resources Engineering, Shanxi Key Laboratory of Geotechnical and Underground Space Engineering XAUAT, Xi'an 710055, China

\* Correspondence: zengxiang_lu@sohu.com (Z.L.); saisai.wu@xauat.edu.cn (S.W.)

**Abstract:** Conducting technical and economic evaluations is important for mining investment and mining operation decision-making. Traditional economic evaluation methods rarely address the issue of evaluation reliability and usually require complex calculations to obtain the optimal solution. In this study, the Rosenblueth point estimate method for reliability evaluation of engineering project schemes is introduced. Combined with the cash flow method for economic evaluation of mines, the Rosenblueth point estimate method for evaluating the reliability of mining economy is established. Based on the technical and economic index of the case mine, taking the ore grade as a sensitivity indicator, empirical research on established models and methods was carried out. The results of the economic reliability evaluation and the variation rules obtained using the Rosenblueth point estimate method model were basically consistent with the actual production and operation rules of mining enterprise. The similar results also proved that the proposed model has good applicability and reliability for mining economic evaluation. Using the proposed RPEM economic reliability model, the economic reliability of a certain iron mine in Liaoning Province was calculated to be 99.95, which was a huge improvement compared with the traditional evaluation method. Additionally, the calculation process of the proposed model for economic reliability evaluation is simple and the accuracy is controllable. The economic reliability of the project can be calculated based on changes in sensitivity indicators, and the value range of sensitivity indicators can also be calculated through the required reliability. The obtained results and the proposed evaluation model provide a decision-making basis for mining investment projects and operation management.

**Keywords:** Rosenblueth method; economic benefit; reliability analysis; economic index

## 1. Introduction

In present times, China's national economy has entered a stage of rapid development. Mineral resources, as an important foundation for promoting social development, must be managed with long-term and stable development paths to ensure the economic benefits of mining enterprises [1–3]. With the primary goal of obtaining economic benefits, mining project development and construction require production processes and modes that can control production costs, improve production efficiency, ensure engineering safety, and maximize profits [4–6]. Economic evaluation can improve operations and management and increase economic benefits. Conducting economic evaluation of mining project development, scientifically predicting the cost level of mining production, improving cost prediction accuracy, discovering the rule of cost changes, and identifying factors that affect costs and effectively controlling them can improve the cost management level and core competitiveness of enterprises [7–9]. Therefore, mining enterprises have a strong focus on how to evaluate their economic benefits.

Currently, the widely used economic evaluation methods include the cash flow method, discounted cash flow method, financial internal rate of return analysis method, net present value and breakeven analysis method, etc. [10]. For example, Zhou Ling [11] used the APRP model for economic evaluation of the green mining evaluation index system and proposed suggestions for the improvement of the growth stage of green mining development in a certain region. Wang Quansheng [12] discovered the problems in using the cash flow method for project economic evaluation and mining rights evaluation and proposed suggestions for the improvement of the determination of relevant parameter selection. In order to improve the utilization rate of resources, Shi Qingbao et al. [13] adopted the internal rate of the return analysis method to evaluate the internal rate of return and the development and utilization value of resource development according to the reserves and industrial indicators in a mining rights project. They proposed the methods and measures for optimizing the internal rate of return. Chen Wenjun et al. [14] utilized the method of combining full lifecycle theory with economic benefit evaluation to evaluate the rationality of mine technical renovation and expansion from aspects such as investment cash flow and financial viability. Based on the traditional capital asset pricing model, Xu Yixin et al. [15] introduced the systematic risk correction coefficient, established an improved capital asset pricing model, and used it to evaluate the value of mining rights. Yang Huaimin et al. [16] suggested that dynamic breakeven analysis can consider the time value of funds, and the evaluation results obtained were more reasonable. Dariusz et al. [17] pointed out that when using the breakeven point to calculate the profitability of mines, the sensitivity of factors should be adjusted based on the impact of each indicator on revenue, taking into account the specific situation of the mine. Wen Wei et al. [18] used TOPSIS and grey correlation method to construct an economic risk evaluation model and evaluated the economic risks of different countries from a macro perspective. Li Guoqing et al. [19] used 0–1 Integer Programming to establish the operation plan optimization model, which could be used to optimize the mining plan, and improve the utilization rate of mineral resources. Kong Wenyuan et al. [20] pointed out that considering infrastructure investment when optimizing mining plans can improve economic benefits and make investment plans more reasonable. Based on the four principles of system comprehensiveness, main factors, operability, and scientific objectivity, Yuliduzi Sidike et al. [21] constructed a mining economic competitiveness evaluation index system with four secondary indicators and 18 tertiary indicators. They used principal component analysis to evaluate the mining economic competitiveness of 10 major mining provinces in China and proposed directions for improving the mining economic competitiveness of Heilongjiang Province.

Various existing evaluation methods have played important roles in mining economic evaluation, but in this process, the calculation procedures of each evaluation method may be cumbersome or simple, and the reliability of evaluation results may be high or low. Seeking a mining economic evaluation method that can simplify the evaluation calculation process and improve the reliability of evaluation results can effectively improve the efficiency and accuracy of evaluation work. The Rosenblueth point estimate method [22] (RPEM) is a method of obtaining reliability and failure probability by calculating the mean–variance of the objective function, which is widely used in engineering reliability analysis [23–25]. RPEM only needs to calculate the objective function at the selected weighted points, which is easy to implement and does not require numerical iteration [24]. However, there is currently no precedent for using this method for economic evaluation. In terms of engineering reliability analysis, Yang Kui [26] used RPEM for slope stability analysis of hydropower stations and compared the calculation results with the results obtained by Monte Carlo method. It was found that the probability of slope failure obtained by these two methods was similar. Cai Qian et al. [27] used RPEM for real-time analysis of the probability of earth-rock dam failure, concluding that RPEM can balance data accuracy and efficiency. Yang Yang et al. [28] applied RPEM to the reliability analysis of tunnel bolt support structures and obtained results that were basically consistent with the actual situation on site, improving computational efficiency. Kahe Maryam Sadat et al. [29] used RPEM to evaluate the

uncertainty of groundwater model parameters, believing that it has the advantages of simplicity and efficiency compared to Monte Carlo methods. Using TRIGRS landslide prediction mode and adopting RPEM to solve the uncertainty problem of parameters, Xu Zenghui et al. [30] studied the relationship between rainfall events and landslide occurrence.

Based on the widespread application of RPEM in engineering reliability evaluation [25], the major objective of this study was to introduce RPEM into mining economic evaluation to shorten the cycle of economic evaluation and improve the reliability of evaluation. The development and production processes of mining projects are complex and involve many technical and economic indicators. In order to study the applicability of RPEM in mining economic evaluation, on the basis of establishing an RPEM economic evaluation method, we studied the probability of achieving benefits with different geological grades, conducted empirical research to verify the rationality of the results, and explored the feasibility of using RPEM for mining economic evaluation.

## 2. RPEM Economic Evaluation Method

### 2.1. The Basic Principles of RPEM

RPEM, also known as moment estimation method, is often used to purposefully select points composed of special values under the condition of unclear distribution of random variables. The mean and variance are used to find the first four moments of the functions point by point and to calculate the reliability of the evaluated project. This method has a simple principle, fewer calculations, and does not need to consider the probability distribution of basic input random variables in advance. The processing of related variables is simple, while the requirements of engineering accuracy can be met [31]. The calculation process of RPEM is shown in Figure 1.

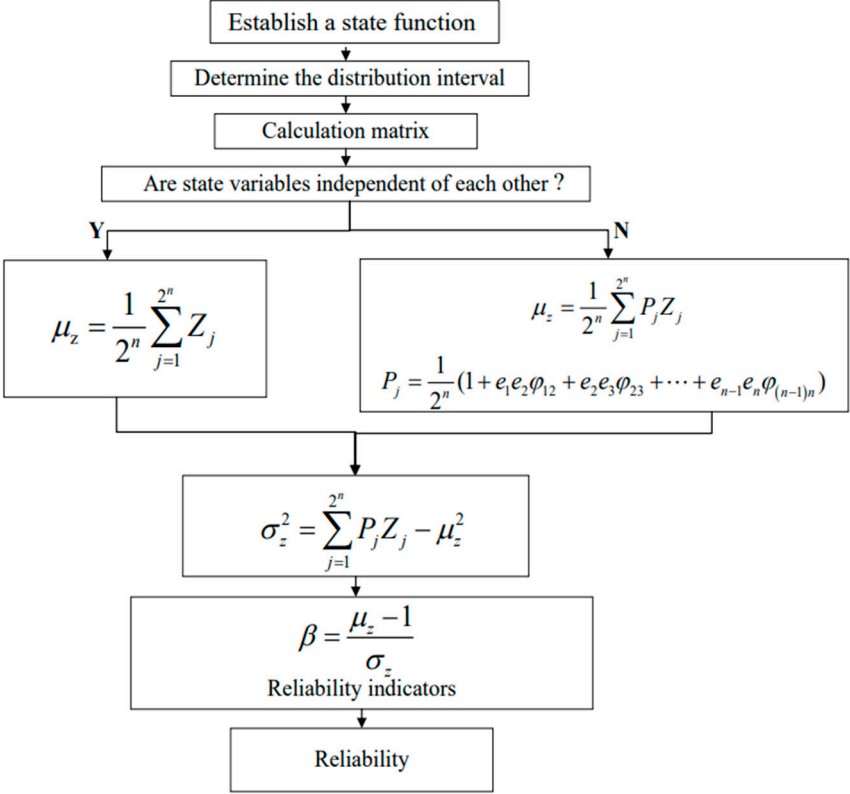

**Figure 1.** Flow chart of RPEM.

When using RPEM for mine technical and economic evaluation, the target state function is first established: $Z = Z (x_1, x_2, \ldots, x_i, \ldots, x_n)$, among which, $x_i$ represents a random variable subject to normal distribution, that is, mining and dressing cost, ore geological grade, concentrate sales price, dressing recovery rate, and other indicators; next, the value points are selected, and within the interval ($x_i$ min, $x_i$ max), two value points are symmetrically seleceted (such as the positive and negative standard deviation of the mean of the state variable). Using $\mu$, $\sigma$ representing the mean and standard deviation of variables, there are:

$$x_{i1} = \mu_{xi} + \sigma_{xi} \tag{1}$$

$$x_{i2} = \mu_{xi} - \sigma_{xi} \tag{2}$$

At this point, there are $2^n$ value points and $2^n$ combinations of corresponding value points for n random variables. When the probabilities of these $2^n$ combinations occurring are equal and independent of each other, the first moment of the mean function is:

$$\mu_z = \frac{1}{2^n} \sum_{j=1}^{2^n} Z_j \tag{3}$$

When a group of variables is related and the probability of occurrence is not equal, the probability of occurrence $P_j$ for each group of variables is related to its correlation coefficient:

$$P_j = \frac{1}{2^n}(1 + e_1 e_2 \varphi_{12} + e_2 e_3 \varphi_{23} + \cdots + e_{(n-1)} e_n \varphi_{(n-1)n}) \tag{4}$$

where $j = 1, 2, 3, \ldots, n$. When $x = x_{i1}$, $e_i = 1$. When $x = x_{i2}$, $e_i = -1$; $\varphi_{ij}$ is the correlation coefficient of a random variable. The first moment of the state function is:

$$\mu_z = \frac{1}{2^n} \sum_{j=1}^{2^n} P_j Z_j \tag{5}$$

The second moment is:

$$\sigma_z^2 = \sum_{j=1}^{2^n} P_j Z_j^2 - \mu_z^2 \tag{6}$$

Reliability indicators $\beta$ can be established as shown in Equation (7):

$$\beta = \frac{\mu_z - 1}{\sigma_z} \tag{7}$$

Then, the calculation of reliability $R$ can be performed using:

$$R = \phi(\beta) \tag{8}$$

*2.2. Construction of RPEM Economic Evaluation Model*

2.2.1. Evaluation Index System

There are numerous and complex indicators that affect the economy of mining projects [32], which can be divided into three secondary indicators: reserve indicators, production indicators, and operational indicators. Then, the indicators are classified according to their attributes to form a three-level indicator. The established technical and economic evaluation index system is shown in Figure 2.

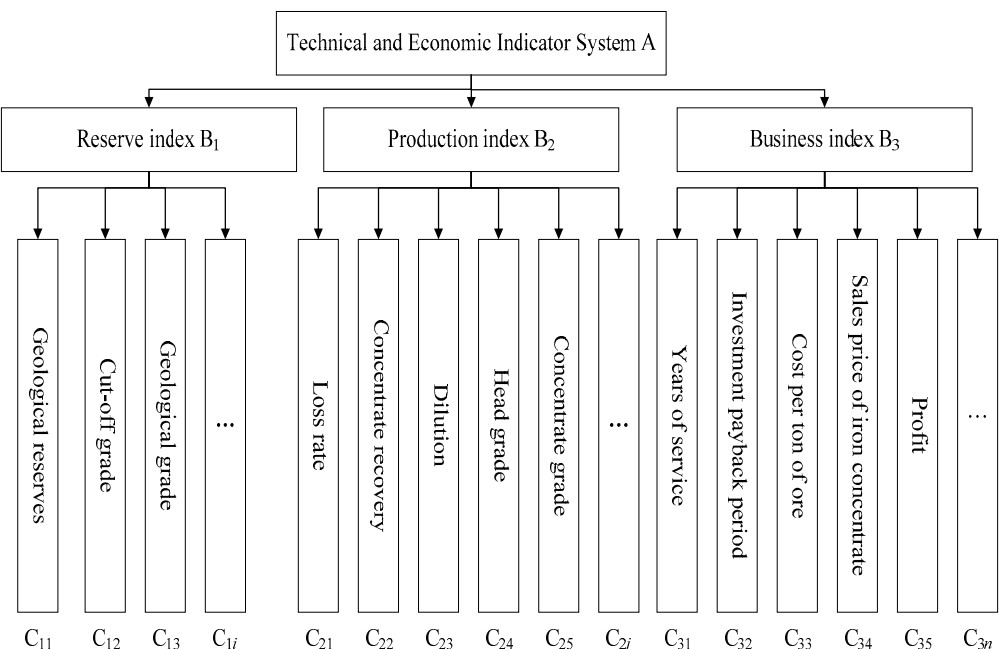

**Figure 2.** Technical and economic index system.

2.2.2. Solving the Reliability of RPEM Economic Evaluation

To study the applicability of RPEM to the economic evaluation of mining, taking the mining and selection production process of underground mines as an example, the commonly used technical and economic indicators are selected for mining economic evaluation and are used to establish a reliability model for economic evaluation of RPEM. The main indicators include mining cost, beneficiation cost, ore dilution rate, ore grade, concentrate grade, beneficiation recovery rate, concentrate sales price, etc. These indicators are all included in the indicator system shown in Figure 2.

When establishing an RPEM economic evaluation reliability model based on the cash flow method, only the costs and benefits directly generated during the mining production process are considered, and the impact of its fluctuations on costs and profits is analyzed using the ore grade as a sensitivity indicator. When the mining and selection production balance and concentrate production and sales balance are achieved, the specific calculations are as follows:

(1) Cash inflows from mines (only when calculating concentrate sales revenue) *F*:

$$F = A \cdot \frac{g \cdot \varepsilon \cdot W}{\gamma} \tag{9}$$

where A is the annual mining output of the mine, 10,000 t/year; G is the ore grade (i.e., the ore grade after beneficiation), %; $\varepsilon$ is the recovery rate of mineral processing, %; *W* is the sales price of concentrate, CNY/t; $\gamma$ is the concentrate grade, %.

(2) Annual cash outflow from mines *P*:

When the production and sales of concentrate produced by the mine are balanced, the annual cash outflow *P* of the mine is:

$$P = A \cdot \frac{g \cdot \varepsilon}{\gamma} \cdot C_c + C_1 + C_2 \tag{10}$$

where Cc is the production cost of ton of ore concentrate, CNY/t; $C_1$ represents various taxes and fees, in CNY 10,000; $C_2$ refers to the annual average fixed assets investment and current asset investment of the enterprise, CNY 10,000.

(3)    Net cash flow of mines *I*:

$$I = F - P = A \cdot \frac{g \cdot \varepsilon}{\gamma} \cdot (W - C_c) - C_1 - C_2 \tag{11}$$

(4)    Construction of Objective State Function for Economic Reliability Evaluation

Based on the basic principles of RPEM, the cash flow method is combined with RPEM to construct the objective state function of the RPEM economic reliability evaluation model, as shown in Equation (12).

$$Z = \frac{F}{P} = \frac{A \cdot g \cdot \varepsilon \cdot W}{A \cdot g \cdot \varepsilon C_c + \gamma(C_1 + C_2)} \tag{12}$$

The economic reliability evaluation target state function represented by Equation (12) reflects the level of benefits that can be achieved by mining projects. The higher the economic reliability, the higher the mining benefits, and vice versa.

From Equation (12), it can also be seen that when the $Z$ value is 1, it indicates that the cash inflow and outflow of the project are equal. At this point, the breakeven point of a certain technical and economic indicator can be determined by formula transformation.

(5)    Solution of Economic Reliability Based on RPEM

First, Equation (12) is used as the objective state function, according to the RPEM calculation process shown in Figure 1; the sensitivity indicators for evaluation and their possible range of values are determined. Secondly, within the range of sensitivity indicators, the mean $\mu z$ and variance $\sigma_z{}^2$ of sensitivity indicators is calculated according to Equations (1)–(6), and Equation (7) is reused to solve reliability index $\beta$. Finally, the reliability R is calculated according to Equation (7). By solving the reliability R under different values of sensitivity indicators and comparing them, the economic reliability of the project under these sensitivity indicators can be analyzed.

The range of values for the economic reliability R obtained from this is 0–1. The larger the R value, the higher the economic reliability, which indicates that the project is more reliable.

## 3. Empirical Study

### 3.1. Case Mine Overview

A certain iron mine in Liaoning Province, China was selected as an example for conducting applied research. The mining area is located in the Liaodong region, with a flat terrain and an inclination angle of 62° to 87° for the ore body, with a thickness of 6.0 to 43.2 meters. The ore body occurs at an elevation between +200 m and −300 m. The main minable ore bodies are Fe10, Fe11-1, and Fe11-2. The main useful mineral is magnetite with a TFe content of 25~35%, which belongs to magnetite lean ore.

The mining of this deposit adopted open-pit multi-step mining operation and a highway transportation method in the early stage. After nearly a decade of open-pit mining, the service life of open-pit mining was only 2.38 years due to the combined constraints of the comprehensive cost of deposit mining and surface terrain conditions. In order to ensure the production efficiency of the enterprise and achieve the full exploitation and utilization of the mining resources, according to the provisions of the "Code for Geological Exploration of Iron, Manganese, and Chromium Mines" (DZ/T0200-2002), the retained reserves of the mine were evaluated. Based on the resource estimation results, as of the evaluation date, the three main minable ore bodies had a total reserve of 17,178.39 kt (122 b + 333), of which the ore bodies below +70 m were mined underground, and the usable reserve of ore was 13,243.82 kt. The mining and beneficiation production capacity of the mine was 1300 kt/a. The main production technical and economic indicators of iron mine in Liaoning Province are shown in Table 1 [33].

**Table 1.** The main production technical and economic indicators.

| Index | Symbol | Underground Mining |
|---|---|---|
| Average geological grade of recoverable reserves (%) | g | 26.3% |
| Dip angle of ore body (°) | α | 62~87 |
| Thickness of ore body (m) | m | 20~40 |
| Actual mining recovery rate (%) | η | 88 |
| Actual dilution rate in mining (%) | ρ | 20 |
| Raw ore (ore extraction) grade (%) | g′ | 21.09 |
| Concentrate grade (%) | γ | 65 |
| Beneficiation recovery (%) | ε | 77.2 |
| Concentrate yield (t/t) | K = G·ε | 0.25 |
| Sales price of iron concentrate (CNY/t) | W | 744 |
| Underground mining cost (CNY/t) | Cp | 78.31 |
| Mineral processing cost (CNY/t) | Cu | 36.28 |
| Production cost of ton of ore concentrate (CNY) | Cc | 114.59 |

*3.2. Empirical Study on the Reliability of RPEM Economic Evaluation*

3.2.1. Determine Model Variables

Using ore grade as a sensitivity indicator, its impact on economic benefits is studied and a state function with ore grade as a variable is established, as shown in Equation (13).

$$Z = \frac{F}{P} = Z(g) \tag{13}$$

3.2.2. Determine the Range of Ore Extraction Grade Values for the Mine

The average geological grade of the mine is 26.3% (as shown in Table 1). When using the non-pillar sublevel caving method for mining, if the ore dilution rate fluctuates between 15% and 20%, the change in the average ore grade g of the mine can be calculated by Equation (14):

$$\begin{aligned} g \ &= g_1 \cdot (1 - \rho) \\ &= 26.3\% \times [1 - (15\% \sim 20\%)] \\ &= 21.04\% \sim 22.36\% \end{aligned} \tag{14}$$

In the formula, $g_1$ is the average geological grade of the deposit, %; ρ is the ore dilution rate, %.

Therefore, when the mine adopts the non-pillar sublevel caving method for mining, the variation range of ore grade in the mine is 21.04~22.36%.

3.2.3. Mine Cash Flow Calculation

According to Equation (14), the variation range of mining grade is obtained, and the economic benefits within this range are discussed. The coefficient of variation of the extracted ore grade is 0.1, and 11 sets of data are taken within the variation range of the ore grade (21.04%, 22.36%). Based on the mining and beneficiation production capacity of the mine, if other indicators remain unchanged, the corresponding concentrate output and concentrates sales revenue for the 11 sets of ore grades can be calculated.

The designed annual mine output of the underground mining project is 1300 kt, the mine ore grade is 21.04%, the production cost of concentrate per ton of ore is CNY 114.59 (excluding tax), and the annual average fixed assets investment and current asset investment is CNY 9.9686 million. The taxes and fees paid by enterprises are subject to changes in sales revenue, as changes in ore grade affect concentrate production, which in turn affects changes in sales revenue. When the ore grade is 21.04%, the annual tax paid by the enterprise is CNY 39.6797 million. Taking a mining grade of 21.04% as an example, the annual cash flow of the mine is calculated, and the specific steps are as follows:

(1)    Annual cash inflows from mines $F$:

$$F = A\frac{g \cdot \varepsilon \cdot w}{\gamma} \quad = 130 \times \frac{21.04\% \times 77.2\% \times 744}{65\%}$$
$$= 241.6941 \, (\text{million Yuan}) \tag{15}$$

(2)    Annual cash outflow from mines $P$:

$$\begin{aligned} P &= A \cdot \frac{g \cdot \varepsilon}{\gamma} \cdot C_c + C_1 + C_2 \\ &= 130 \times \frac{21.04\% \times 77.2\%}{65\%} \times 114.59 + 3967.97 + 996.86 \\ &= 198.6153 \, (\text{million Yuan}) \end{aligned} \tag{16}$$

(3)    Annual net cash flow of the mine $I$:

$$I = F - P \quad = 241.6941 - 198.6153$$
$$= 43.0788 \, (\text{million Yuan}) \tag{17}$$

3.2.4. Calculation of Economic Reliability of Mines

(1)    Establish target state function

Taking the ore extraction grade of 21.04% as an example and referring to Equation (13), a target state function with the ore extraction grade as a variable is established, as shown in Equation (18):

$$Z = \frac{F}{P} = Z(21.04\%) \tag{18}$$

(2)    Solution to Economic Reliability

According to the basic principles and methods of RPEM in Section 1, with its coefficient of variation of 0.1, two symmetrical value points for an ore grade of 21.04% can be obtained according to Equations (1) and (2),

$$x_{i1} = \mu_{xi} + \sigma_{xi} = 21.04\% + 2.1\% = 23.14\% \tag{19}$$

$$x_{i2} = \mu_{xi} - \sigma_{xi} = 21.04\% - 2.1\% = 18.94\% \tag{20}$$

By using the cash flow calculation method for mines, the cash flows for ore grades of 23.14% and 18.94% are calculated, resulting in $Z_1 = 1.2814$ and $Z_2 = 1.1464$. Its mean $\mu_z$ and variance $\sigma_z^2$ of the state function for ore grade variables within the interval of 21.04% and 22.36% can be calculated, as shown in Equations (21) and (22):

$$\mu_z = \frac{1}{2}(Z_1 + Z_2) = \frac{1}{2} \times (1.2814 + 1.1464) = 1.2139 \tag{21}$$

$$\sigma_z^2 = \sum_{j=1}^{2^n} P_j Z_j^2 - \mu_z^2 = 0.0091 \tag{22}$$

According to the calculation results of Equations (21) and (22), Equation (7) is used to solve the reliability index of the state function $\beta$, as shown in Equation (23).

$$\beta = \frac{\mu_z - 1}{\sigma_z} = \frac{0.2139}{0.0954} = 2.2421 \tag{23}$$

At this point, when the ore extraction grade is 21.04%, the economic reliability $R$ of the mine is:

$$R = \phi(2.2421) = 98.75\% \tag{24}$$

Within the variation range of ore grade [21.04%, 22.36%], with a coefficient of variation of 0.1, taking 11 sets of data, the cash flow, reliability indicators, and economic reliability of

each group of data are calculated separately according to the above steps. The results are shown in Table 2.

**Table 2.** Reliability calculation results.

| Ore Grade /% | Concentrate Production /t | Income /CNY 10,000 | Net Cash Flow /CNY 10,000 | Reliability Indicators | Economic Reliability /% |
|---|---|---|---|---|---|
| 21.04 | 4.21 × 105 | 2.42 × 104 | 0.43 × 104 | 2.2421 | 98.75 |
| 21.17 | 4.23 × 105 | 2.43 × 104 | 0.44 × 104 | 2.2759 | 98.84 |
| 21.30 | 4.26 × 105 | 2.45 × 104 | 0.45 × 104 | 2.3226 | 98.98 |
| 21.43 | 4.29 × 105 | 2.46 × 104 | 0.46 × 104 | 2.3647 | 99.09 |
| 21.57 | 4.31 × 105 | 2.48 × 104 | 0.47 × 104 | 2.4031 | 99.18 |
| 21.70 | 4.34 × 105 | 2.49 × 104 | 0.48 × 104 | 2.4473 | 99.27 |
| 21.83 | 4.37 × 105 | 2.51 × 104 | 0.49 × 104 | 2.4924 | 99.36 |
| 21.96 | 4.33 × 105 | 2.52 × 104 | 0.50 × 104 | 2.5292 | 99.41 |
| 22.09 | 4.42 × 105 | 2.54 × 104 | 0.51 × 104 | 2.5682 | 99.49 |
| 22.22 | 4.44 × 105 | 2.55 × 104 | 0.52 × 104 | 2.6115 | 99.55 |

Repeating the above process, the cash flow, reliability indicators, and economic reliability of its variables when they change within its value range can be obtained.

### 3.3. Analysis of Calculation Results

According to Table 2, when the mining grade increased from 21.04% to 22.36%, the company's revenue increased from CNY 2.42 × 104 ten thousand to CNY 2.57 × 104 ten thousand, an increase of 10.33%; net cash flow increased by CNY 0.1 × 104 ten thousand, an increase of 23.26%; and the economic reliability calculated using the RPEM-based economic reliability model increased from 98.75% to 99.95%, an increase of 1.2%.

It can be seen that as the ore grade increases, the concentrate production increases, and the profits and net cash flow of the mine also increase. The economic reliability calculated based on the RPEM economic reliability evaluation model is also continuously increasing. This feature is completely consistent with the actual production and operation situation of mining enterprises. This indicates that the economic reliability evaluation model established based on RPEM has its applicability for the economic evaluation of mining projects.

To more intuitively predict the trend of changes in reliability and revenue, the annual revenue and economic reliability data of enterprises in Table 2 are converted into a curve chart, as shown in Figure 3.

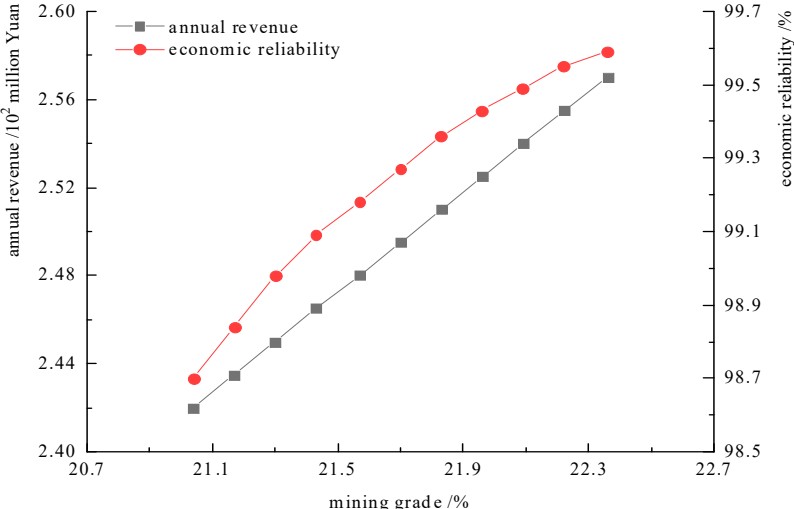

**Figure 3.** Relationship curve between annual revenue and economic reliability with changes in ore grade.

From Figure 3, it can be seen that as the ore grade increases, the mine income curve and economic reliability curve also change and increase, indicating that while the enterprise income increases, the economic reliability also increases. The trend of curve changes is in line with the objective facts of mine production and operation.

The empirical research results indicate that the mining economic reliability evaluation model established based on the RPEM method has good applicability, and the evaluation results comply with the basic laws of changes in mining production and operation.

## 4. Conclusions

The RPEM method for studying the economic reliability of mines was introduced. Combined with the cash flow method, a reliability model for mining economic evaluation was proposed. Based on the actual production of empirical mines and the impact of dilution rate on ore grade, the impact of changes in ore grade on mine revenue and economic reliability was studied using ore grade as a sensitivity indicator. The established RPEM economic reliability evaluation model was validated. The empirical research results indicate that the economic reliability results and their variation patterns obtained by using the RPEM evaluation model are basically consistent with the actual prodHanuction and operation laws of mining enterprises, indicating that the established RPEM economic evaluation method has good applicability and reliability for mining economic evaluation. By analyzing the changes in economic reliability when different sensitive technical and economic indicators change, the problem of mining economic evaluation can be solved and the reasonable range of relevant parameters can be quickly found. The RPEM calculation process is simple and can solve the economic reliability problem of the project based on changes in sensitivity indicators, and can also calculate the value range of sensitivity indicators based on the required reliability. The obtained results and the proposed evaluation model provide a decision-making basis for the production and operation of mining enterprises. Future study involving the dynamic evaluation indicators in mining economic evaluation, expenditures and profits outside the mining production enterprises is also suggested.

**Author Contributions:** Conceptualization and methodology, J.L.; validation and formal analysis, T.W.; resources, data curation and supervision, Z.L.; writing—review and editing, S.W. All authors have read and agreed to the published version of the manuscript.

**Funding:** The work was supported by the National Nature Science Foundation of China (No. 51774176), Shannxi Province Key Research and Development Program (2023-GHYB-06).

**Institutional Review Board Statement:** Not applicable.

**Informed Consent Statement:** Not applicable.

**Data Availability Statement:** The detailed data is available upon request.

**Acknowledgments:** Not applicable.

**Conflicts of Interest:** The authors declare no conflict of interest.

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
