# Peer review of "Investigation into Mining Economic Evaluation Approaches Based on the Rosenblueth Point Estimate Method"

_applsci, doi:10.3390/app13159011_

Round 1

Reviewer 1 Report

(1) The abstract and conclusions of the manuscript needs to include some key quantitative data. In this way, it can be more conducive to revealing the results obtained by the author.

(2) What is the innovation of the RPEM method mentioned in the manuscript, and why do the authors use this method? Does its applicability also need to be effectively verified?

(3) What are the sources of data provided for research in Table 1? Please clarify the source of the data in the table.

(4) How much economic reliability can meet the economic requirements and goals of mining? Is there any basis at present?

(5) Some references needs to be cited to support the statement in Lines 40-42 that With the primary goal of obtaining economic benefits, mining project development and construction strive to pursue production processes and modes that can control production costs, improve production efficiency, ensure engineering safety, and maximize profits. â‘  https://doi.org/10.1016/j.oceaneng.2023.114949; â‘¡ https://doi.org/10.1007/s11053-023-10202-7; â‘¢ https://doi.org/10.1016/j.jcis.2022.12.160.

(6) From the model provided in the manuscript, it can be observed that there are many factors that affect annual income. However, in Figure 3, with the improvement of mining grade, the annual income shows a linear growth? In addition, in Figure 3, the economic reliability gradually approaches 100%, but cannot reach 100%. Why is this? In other words, what is the reason why economic reliability slows down as mining grade increase?

The language of the manuscript is basically acceptable, and the author only needs to check for minor grammar issues.

Author Response

*** Please note that the referenced line numbers refer to the track-changed manuscript.

Reviewer 1

The authors would like to thank the reviewer for the constructive comments in improving the quality of the manuscript. The manuscript has been revised according to the comments.

(1) The abstract and conclusions of the manuscript needs to include some key quantitative data. In this way, it can be more conducive to revealing the results obtained by the author.

Thank the reviewer for the comments. To address the concern of the reviewer, the following contents has been added to the revised manuscript as follows:”Using the proposed RPEM economic reliability model, the economic reliability of a certain iron mine in Liaoning Province was calculated to be 99.95%, which was a huge improvement compared with the traditional evaluation method”. (see lines 25-28)

(2) What is the innovation of the RPEM method mentioned in the manuscript, and why do the authors use this method? Does its applicability also need to be effectively verified?

Thank the reviewer for the comments. As the authors presented in the manuscript that “Various existing evaluation methods have played a good role in mining economic evaluation, but in mining economic evaluation, the calculation procedures of each evaluation method may be cumbersome or simple, and the reliability of evaluation results may be high or low. Seeking a mining economic evaluation method that can simplify the evaluation calculation process and improve the reliability of evaluation results can effectively improve the efficiency and accuracy of evaluation work.” (see lines 87-92)

Based on the knowledge of the authors, the RPEM calculation process for economic reliability evaluation is simple and the accuracy is controllable. Therefore, the major objective of the manuscript is to provide the methodologies for applying the RPEM method in the economic reliability evaluation as well as verify the efficiency using the data of a certain iron mine in Liaoning Province. To address the concern of the reviewer, the following contents has been added to the revised manuscript as follows:” the major objective of the study is introducing the RPEM into mining economic evaluation to shorten the cycle of economic evaluation and improve the reliability of evaluation”. (see lines 112-114)

(3) What are the sources of data provided for research in Table 1? Please clarify the source of the data in the table.

The sources of data provided in Table 1 are from the monograph of one of the authors. To address the concern of the reviewer, The sources of data provided for research in Table 1 has been clarified and cited in the revised manuscript as follows:” The main production technical and economic indicators of iron mine in Liaoning Province are shown in Table 1. [27]” (see lines 244-246)

(4) How much economic reliability can meet the economic requirements and goals of mining? Is there any basis at present?

Thank the reviewer for the comments. The economic reliability to meet the economic requirements is the ultimate objective for this study. For the best knowledge of all the authors, there is no such a standard for the minimum requirement of the economic evaluation reliability. Therefore, in the real practice, the higher of the evaluation reliability, the better. In this study, using the proposed method, the economic reliability of a certain iron mine in Liaoning Province was calculated to be 99.95%,

(5) Some references needs to be cited to support the statement in Lines 40-42 that With the primary goal of obtaining economic benefits, mining project development and construction strive to pursue production processes and modes that can control production costs, improve production efficiency, ensure engineering safety, and maximize profits.â‘ https://doi.org/10.1016/j.oceaneng.2023.114949; â‘¡ https://doi.org/10.1007/s11053-023-10202-7; â‘¢ https://doi.org/10.1016/j.jcis.2022.12.160.

Thanks for the reviewer for providing the useful information. To make a more comprehensive literature review, the suggested references have been cited.

(6) From the model provided in the manuscript, it can be observed that there are many factors that affect annual income. However, in Figure 3, with the improvement of mining grade, the annual income shows a linear growth? In addition, in Figure 3, the economic reliability gradually approaches 100%, but cannot reach 100%. Why is this? In other words, what is the reason why economic reliability slows down as mining grade increase?

Thank the reviewer for the comments. In the real practice, the development and production processes of mining projects are complex and involve many technical and economic indicators. For example, the occurrence of any potential risk may have a huge effect on the economic development. That is also one of the reasons for conducting the economic evaluation reliability analysis in this study. Generally, the more factors and data involved, the more accuracy of the reliability, but it will not reach 100% since too many factors including some factors that unknown during the assessment process. The reason why economic reliability slows down as mining grade increase is because the resources recovery rate of the mineral resources. In other words, the largest recovery rate of the mine around the world is around 90%. The higher the grade, the higher effects of the resources recovery rate on the income of the mine as well as the economic reliability.

Reviewer 2 Report

A reasonably reliable and accurate RPEM method for studying the economic reliability of mines was introduced.

Application of the method allowed to receive provided decision-making basis for the production and operation of mining enterprise

However, the presented material does not provide an opportunity to assess whether this method is better than those presented in detail in the review. There is no comparison of methods even on simple examples

Author Response

*** Please note that the referenced line numbers refer to the track-changed manuscript.

Reviewer 2

The authors would like to thank the reviewer for the constructive comments in improving the quality of the manuscript. The manuscript has been revised according to the comments.

A reasonably reliable and accurate RPEM method for studying the economic reliability of mines was introduced.

Thanks for the reviewer for the comments.

Application of the method allowed to receive provided decision-making basis for the production and operation of mining enterprise.

Thanks for the reviewer for the comments.

However, the presented material does not provide an opportunity to assess whether this method is better than those presented in detail in the review. There is no comparison of methods even on simple examples

Thank the reviewer for the comments. Actually, the comparison of the proposed method and the traditional ones are quite important for this study. The authors would like to kindly explain that in the review parts, some of the traditional methods are provided (see lines 51-86) and the efficiency of the proposed is verified using an example of a certain iron mine in Liaoning Province.

Thanks for your valuable comments again.

Reviewer 3 Report

The submitted manuscript entitled: «Investigation into the mining economic evaluation approaches based on Rosenblueth point estimate method»

is interesting and in a scientific way develops the economic evaluation approaches using the specific method.

The following points need to be clarified and revised.

1.       Keywords should not repeat words in the title. So any common words should be deleted and replaced by other words.

2.       Lines 12-13. The sentence seems incomplete. It is the first sentence of the text and should be clear and complete.

3.       The abstract needs to be supplemented with quantitative data, which have been derived from this work.

4.       Lines 108-110. The sentence is long and the question is vague.

5.       «2.2.2 Solving the Reliability of RPEM Economic Evaluation» Please explain why only the costs and benefits directly generated during the mining production process are considered while establishing an RPEM economic evaluation reliability model. The approach and concept should be based on current literature.

6.       The conclusions are well written, however, suggestions for future research should be highlighted as a follow-up to this study.

7.       The whole manuscript needs strengthening with concurrent and additional references.

8.       Language corrections need to be made by a native English speaker throughout the text.

Moderate editing of English language required

Author Response

*** Please note that the referenced line numbers refer to the track-changed manuscript.

Reviewer 3

The authors would like to thank the reviewer for the constructive comments in improving the quality of the manuscript. The manuscript has been revised according to the comments.

The submitted manuscript entitled: «Investigation into the mining economic evaluation approaches based on Rosenblueth point estimate method» is interesting and in a scientific way develops the economic evaluation approaches using the specific method.

The following points need to be clarified and revised.

  1. Keywords should not repeat words in the title. So any common words should be deleted and replaced by other words.

Thank the reviewer for the suggestion and the Keywords have been adjusted as follows:” Rosenblueth method; economic benefit; reliability analysis; economic index”. (see line 34)

  1. Lines 12-13. The sentence seems incomplete. It is the first sentence of the text and should be clear and complete.

Thank the reviewer for the suggestion, the line 12-13 has been revised as follows:” Conducting technical and economic evaluations is quite important for mining investment and mining operation decision-making”. (see line 12-13)

  1. The abstract needs to be supplemented with quantitative data, which have been derived from this work.

Thank the reviewer for the comments. To address the concern of the reviewer, the following contents has been added to the revised manuscript as follows:”Using the proposed RPEM economic reliability model, the economic reliability of a certain iron mine in Liaoning Province was calculated to be 99.95%, which was a huge improvement compared with the traditional evaluation method”. (see lines 25-28)

  1. Lines 108-110. The sentence is long and the question is vague.

Thank the reviewer for the suggestion, the line 108-110 has been revised as follows:” Based on the widespread application of RPEM in engineering reliability evaluation [24], the major objective of the study is introducing the RPEM into mining economic evaluation to shorten the cycle of economic evaluation and improve the reliability of evaluation”. (see lines 112-114)

  1. «2.2.2 Solving the Reliability of RPEM Economic Evaluation» Please explain why only the costs and benefits directly generated during the mining production process are considered while establishing an RPEM economic evaluation reliability model. The approach and concept should be based on current literature.

In the real practice, the development and production processes of mining projects are complex and involve many technical and economic indicators. For example, the occurrence of any potential risk may have a huge effect on the economic development. That is also one of the reasons for conducting the economic evaluation reliability analysis in this study. Generally, the more factors and data involved, the more accuracy of the reliability. Additionally, the study find that economic reliability slows down as mining grade increase. The reason why economic reliability slows down as mining grade increase is because the resources recovery rate of the mineral resources. In other words, the largest recovery rate of the mine around the world is around 90%. The higher the grade, the higher effects of the resources recovery rate on the income of the mine as well as the economic reliability. Therefore, the grade, costs and benefits and others factors related to the income are involved.

  1. The conclusions are well written, however, suggestions for future research should be highlighted as a follow-up to this study.

Thank the reviewer for the suggestion and suggestions for future research has been highlighted as follows:” Future study involving the dynamic evaluation indicators in mining economic evaluation, expenditures and profits outside the mining production enterprises is also suggested.” (see lines 361-362)

  1. The whole manuscript needs strengthening with concurrent and additional references.

Thank the reviewer for the suggestion. To make a more comprehensive literature review, more references and contents have been added.

  1. Language corrections need to be made by a native English speaker throughout the text.

Thank the reviewer for the suggestion, the manuscript has been reviewed by an native English-speaking editor, and the following adjustments have been made. (see lines 11, 25-27, 101-103, 113-114, 120-121, 126, 128……)

Round 2

Reviewer 1 Report

At present, the quality of the manuscript can fully meet the standards for acceptance and acceptance. Therefore, it is recommended that the editorial department give it a decision to accept.

Reviewer 3 Report

The revised manuscript now fulfills the requirements for publication in the journal.